# The Disproportionate Rise in Pancreatic Cancer in Younger Women Is Due to a Rise in Adenocarcinoma and Not Neuroendocrine Tumors: A Nationwide Time-Trend Analysis Using 2001–2018 United States Cancer Statistics Databases

**DOI:** 10.3390/cancers16050971

**Published:** 2024-02-28

**Authors:** Yi Jiang, Yazan Abboud, Jeff Liang, Brent Larson, Arsen Osipov, Jun Gong, Andrew E. Hendifar, Katelyn Atkins, Quin Liu, Nicholas N. Nissen, Debiao Li, Stephen J. Pandol, Simon K. Lo, Srinivas Gaddam

**Affiliations:** 1Karsh Division of Gastroenterology, Departments of Medicine, Cedars-Sinai Medical Center, Los Angeles, CA 90048, USA; yazanabboud.md@gmail.com (Y.A.); jeff.liang@cshs.org (J.L.); quin.liu@cshs.org (Q.L.); stephen.pandol@cshs.org (S.J.P.); simon.lo@cshs.org (S.K.L.); 2Department of Pathology, Cedars-Sinai Medical Center, Los Angeles, CA 90048, USA; 3Division of Medical Oncology, Cedars-Sinai Medical Center, Los Angeles, CA 90048, USA; arsen.osipov@cshs.org (A.O.); jun.gong@cshs.org (J.G.); andrew.hendifar@cshs.org (A.E.H.); 4Department of Radiation Oncology, Cedars-Sinai Medical Center, Los Angeles, CA 90048, USA; katelyn.atkins@cshs.org; 5Department of Surgery, Cedars-Sinai Medical Center, Los Angeles, CA 90048, USA; nicholas.nissen@cshs.org; 6Biomedical Imaging Research Institute, Cedars Sinai Medical Center, Los Angeles, CA 90048, USA; debiao.li@cshs.org

**Keywords:** pancreatic ductal adenocarcinoma, pancreatic neuroendocrine tumors, sex disparity

## Abstract

**Simple Summary:**

Previous studies have noted a significant increase in the incidence of pancreatic cancer among younger women (15–54 years) compared to men in the United States, yet the specific histopathologic types remained unexplored. This study aimed to elucidate whether the disproportionate rise in the incidence of pancreatic cancer in younger women was predominantly attributed to pancreatic ductal adenocarcinoma (PDAC) or pancreatic neuroendocrine tumors (PanNET). Our analysis revealed that the age-adjusted incidence rates (aIRs) of PDAC in younger women increased at a greater rate than counterpart men, whereas PanNET did not demonstrate a statistically significant sex-specific average annual percentage change difference. These discoveries provide crucial insights for guiding future investigations and informing healthcare policy.

**Abstract:**

In previous studies, a significant increase in the incidence of pancreatic cancer among younger women compared to men in the United States was noted. However, the specific histopathologic characteristics were not delineated. This population-based study aimed to assess whether this disproportionate rise in pancreatic cancer in younger women was contributed by pancreatic ductal adenocarcinoma (PDAC) or pancreatic neuroendocrine tumors (PanNET). The United States Cancer Statistics (USCS) database was used to identify patients with pancreatic cancer between 2001 and 2018. The results showed that, in younger adults, the incidence of PDAC has increased in women [average annual percentage change (AAPC) = 0.62%], while it has remained stable in men (AAPC = −0.09%). The PDAC incidence rate among women increased at a greater rate compared to men with a statistically significant difference in AAPC (*p* < 0.001), with neither identical nor parallel trends. In contrast, cases of PanNET did not demonstrate a statistically significant sex-specific AAPC difference. In conclusion, this study demonstrated that the dramatic increase in the incidence rate of PDAC explains the disproportionate rise in pancreatic cancer incidence in younger women. This prompts further prospective studies to investigate the underlying reasons for these sex-specific disparities in PDAC.

## 1. Introduction

Globally, pancreatic cancer has become the third leading cause of cancer-related deaths, with a rising incidence [1]. More than 95% of pancreatic cancers arise from the exocrine elements, with pancreatic ductal adenocarcinoma (PDAC) accounting for the majority of pancreatic neoplasms [2]. Pancreatic neuroendocrine tumors (PanNET), arising from the endocrine pancreas, comprise less than 5% of pancreatic neoplasms. A previous study analyzing the Surveillance, Epidemiology, and End Results Program (SEER) database showed a greater relative increase in the incidence of pancreatic cancer among women younger than the age of 55 compared to counterpart men [3]. Despite the concerning findings, little is known regarding the histopathological subtype of pancreatic cancer that is contributing to this rapid rise in incidence among younger women. This study aimed to perform an age and sex-specific time–trend analysis of the age-adjusted incidence rates (aIRs) of PDAC and PanNET using the United States Cancer Statistics (USCS) database (Available online: www.cdc.gov/cancer/uscs/public-use (accessed on 23 October 2021).), which represents nearly 100% of the US population [4]. 

## 2. Materials and Methods

This is a retrospective analysis of prospectively collected data in the National Program of Cancer Registries (NPCR) [5] and the Surveillance, Epidemiology, and End Results Program (SEER) [6], which comprise the United States Cancer Statistics (USCS) databases. SEER*stat software [version 8.3.9.2, National Cancer Institute (NCI), Bethesda, MD] was used to retrieve cases of cancer in which the pancreas was the primary site and that were diagnosed between 2001 to 2018 using the International Classification of Diseases-Oncology-3 (ICD-O-3) codes (Appendix B) [7]. Only cases with malignant behavior were included. The primary outcomes were the PDAC and PanNET sex-specific trends and age-adjusted incidence rates (aIRs) per 100,000 population among multiple age-specified groups. This database did not contain any protected health information. Given that both the SEER and NPCR databases are de-identified and publicly available, institutional review board approval was not required.

To investigate the robustness of our findings, sensitivity analyses were performed to examine the age and sex-specific time trends of the aIRs of PDAC and PanNET exclusively in cases with microscopically confirmed diagnosis. The sex-specific trends and aIRs among multiple histology-specified groups in microscopically confirmed younger adult (15–54 years) PDAC cases were also analyzed. In some situations, PDAC and PanNET are difficult to distinguish, particularly in the poorly differentiated or undifferentiated subgroups [8]. We performed a sensitivity analysis while excluding the poorly and undifferentiated subgroups, using restricted PDAC and PanNET ICD-O-3 codes (Appendix B). 

For age-specific analyses, we divided the age groups into 15–54 years of age (younger adults) and more than or equal to 55 years of age (older adults). The younger adults were further categorized into two equal-age subgroups. This subsequent analysis was performed in the 35–54 years and 15–34 years age groups.

The joinpoint regression program (v4.9.0.1, NCI, Bethesda, MD, USA) was used to assess trends. AIRs were calculated per 100,000 population (age-adjusted to the US population in the year 2000). The annual percentage change and average annual percentage change (APC and AAPC) were calculated. The quantification of time trends was performed using Monte Carlo permutation analysis to fit the simplest joinpoint model. Pairwise comparison was used to assess identicalness and parallelism. Statistical significance was defined as a 2-sided p-value cut-off of 0.05. *p*-value < 0.025 was considered statistically significant for the post hoc age subgroups.

## 3. Results

Between 2001 and 2018, 748,132 patients were diagnosed with pancreatic cancer (aIR 12.35/100,000). Of these, 694,661 patients had PDAC (92.9% of all pancreatic cancer cases; 49.5% women; aIR 11.46/100,000). The >55 year-old age group represented 89.8% and the 15–54 year-old age group represented 10.2% of all PDAC cases. There were 42,806 patients diagnosed with PanNET (5.7% of all pancreatic cancer cases; 45.3% women; aIR 0.71/100,000). The >55 year-old age group represented 69.6% and the 15–54 year-old age group represented 30.3% of all PanNET cases.

The overall aIRs for PDAC significantly increased in both women and men of all ages and older adults without a significant difference. In younger adults, the aIR of PDAC increased in women [AAPC = 0.62% (95% CI 0.30%–0.94%), *p* < 0.001] but remained stable in men [AAPC = −0.09% (95% CI −0.32%–0.14%), *p* = 0.44]. There was a statistically significant difference in AAPC: −0.71% (95% CI −1.07%–−0.34%, *p* < 0.001). These trends were non-identical (*p* < 0.001) and non-parallel (*p* = 0.02), suggesting that the aIR among women increased at a greater rate than in men. This sex disparity was most significant in the 35–54 year-old age subgroup (Table 1, Figure 1). 

Cases of PanNET did not demonstrate a statistically significant sex-specific AAPC difference. The aIRs of PanNET increased in both women and men in the prespecified age groups, including all ages, older adults, and younger adults (15–34 and 35–54 year-old age subgroups) (Table 2, Figure 2).

The results from the sensitivity analyses were consistent with the primary analysis (Appendix A).

Trend analyses on histologic subgroups were performed on microscopically confirmed PDAC in younger adults (Appendix A). A PDAC subgroup that was not otherwise specified (NOS) (74.6% of all PDAC subtypes) demonstrated sex disparity, in which the aIRs among women [AAPC = 1.02% (95% CI 0.68%–1.37%), *p* < 0.001] increased at a greater rate compared to men [AAPC = 0.37% (95% CI 0.05%–0.68%), *p* = 0.03], with a statistically significant difference of −0.66% (95% CI −1.09%–−0.23%, *p* = 0.003); these were non-identical (*p* < 0.001) trends. Other histologic subtypes (intraductal papillary mucinous neoplasm, adenosquamous carcinoma, medullary carcinoma, signet ring cell carcinoma, cystic adenocarcinoma) were too few to estimate a trend. Mucinous PDAC showed similarly decreasing trends of aIRs in both sex groups (*p* = 0.18).

## 4. Discussion

This study showed a greater increase in aIR in younger adult women compared to counterpart men with PDAC but not with PanNET. These trends were statistically different and non-identical. Further evaluation showed that PDAC, NOS (ICD-O-3 code specified as 8140) was likely contributed to this sex disparity in young adults with PDAC.

The exact cause of this notably increased incidence of PDAC in younger women remains unknown, but is likely linked to imbalanced risk factors [9]. Smoking and tobacco use have been recognized as the primary risk factor for young-onset pancreatic cancer [10,11,12]. According to a Swedish prospective, population-based study, while regular smoking is considered as a risk factor for both sexes, occasional and passive smoking were identified as significant risk factors only for women [13]. With improved smoking cessation interventions, a significant reduction in tobacco use amongst young people has been noted globally, along with a decline in disease-adjusted life years attributed to tobacco use, particularly in younger women [10]. Heavy alcohol consumption is directly linked to an increased risk of pancreatic cancer [9] and indirectly associated with pancreatitis, a well-established risk factor for this disease [14]. Historically, men drank more and experienced more alcohol-related harm; however, a recent trend suggests convergence. A meta-analysis demonstrated a trend towards the closing of the sex gap in alcohol consumption and its associated harms. This finding suggests a rising prevalence of alcohol consumption and its adverse effects in women, especially among the younger cohort born in the late 1990s [15]. Previous publications highlighted the essential role of estrogen-related receptor γ in pancreatic acinar cell function; the suggested reproductive factors may contribute to the onset of pancreatic diseases [16]. Research on the association between reproductive factors and the risk of PDAC has shown heterogenous results. Some studies have suggested that high parity [17,18], having had two children [19], and a cumulative breastfeeding duration of over 24 months [20] are associated with a decreased risk of pancreatic cancer. However, subsequent large cohort studies [21,22] have failed to identify a strong association between reproductive factors and pancreatic cancer risk. Similarly, there are controversial results regarding the association between the use of exogenous hormones and the risk of pancreatic cancer [23,24,25]. Later, it was proposed that polymorphism in estrogen-related genes may contribute to PDAC susceptibility, with a distinct molecular landscape particularly associated with young-onset PDAC [26,27,28]. 

On the other end, the incidence of PanNET was found to be consistently increasing across all age groups without significant sex disparity. These findings are consistent with previous studies [29,30]. It was suspected that the rising incidence of PanNET is largely attributed to the increased detection of asymptomatic disease due to the heightened awareness and the expanded utilization of abdominal imaging [31] and endoscopy [32]. Despite both PanNET and PDAC being associated with potential risk factors such as smoking, heavy alcohol use, and diabetes [33], the patterns of these risk factors and the extent of their impact differ. Further investigation is needed to elucidate the role of PDAC-exclusive risk factors in contributing to the observed sex disparity.

Previous studies suspected that the greater increase in the aIRs of pancreatic cancer in younger women could be attributed to the accelerating incidence of histologic subtypes that are more common in women, such as cystic PDAC and solid pseudopapillary neoplasm (SPN) [34]; however, our study does not show the same findings. SPNs (which occur predominantly in young women) were excluded from our analysis given that they are different from PDAC or PanNET. Our findings further confirm that it is the NOC PDAC subtype that is responsible for the sex disparity.

Given the ICD code-based data extraction, the limitations of this study include known issues with the coding accuracy achievable with large databases, a lack of information on major modifiable risk factors in the database, including obesity, smoking, and alcohol consumption, as well as a lack of information on their sex-specific, age-specific, or region-specific effects. The strengths of this study include a strong statistical methodology, such as testing for identicalness and parallelism. Sensitivity analyses confirmed the robustness of our findings. 

## 5. Conclusions

Using nationwide data representing nearly 100% of the US population, we demonstrated that the disproportionate rise in the incidence of pancreatic cancer in younger women is explained by the PDAC histological subtype. PanNET, which does not share the same pattern of common risk factors with PDAC, appears to be increasing equally in both men and women. This raises further questions about the cause of this sex disparity in PDAC. Further prospective population-based studies with age–period modeling and the adjustment of potential lifestyle-related confounders are needed to explore sex-based disproportional exposure to PDAC-exclusive risk factors.

## Figures and Tables

**Figure 1 cancers-16-00971-f001:**
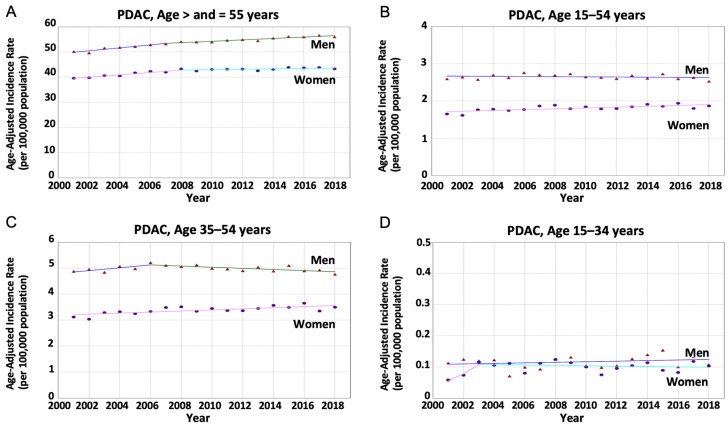
Age-adjusted incidence rates for pancreatic ductal adenocarcinoma (PDAC) stratified by age group and sex. (*Circle*) Incidence rate in women. (*Triangle*) Incidence rate in men. (**A**) The average annual percentage change (AAPC) was greater in men compared to women but not statistically significant (0.73% vs. 0.58%); *p* = 0.001 indicating non-parallel trends among people aged > and =55 years. (**B**) The AAPC was significantly greater in women compared to men (0.62% vs. −0.09%); *p* = 0.02 indicating non-parallel trends among people aged 15–54 years. (**C**) The AAPC was significantly greater in women compared to men (0.62% vs. 0.01%); *p* = 0.003 indicating non-parallel trends among people aged 35–54 years. (**D**) The AAPC increased in women and men without a statistically significant difference (3.36% vs. 0.78%); *p* = 0.04 indicating non-parallel trends among people aged 15–34 years.

**Figure 2 cancers-16-00971-f002:**
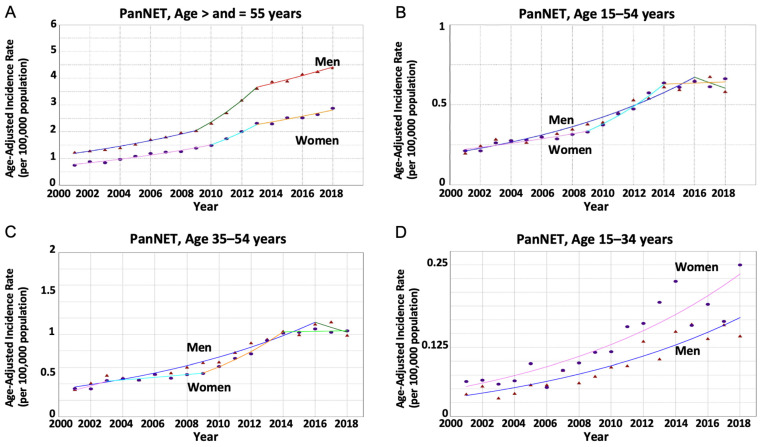
Age-adjusted incidence rates for pancreatic neuroendocrine tumors (PanNET) stratified by age group and sex. (*Circle*) Incidence rate in women. (*Triangle*) Incidence rate in men. (**A**) The average annual percentage change (AAPC) in PanNET was greater in women compared to men but not statistically significant (7.83% vs. 8.0%); *p* = 0.01 indicating non-parallel trends among people aged > and =55 years. (**B**) The AAPC was not significantly different between women and men (6.44% vs. 6.36%); *p* = 0.72 indicating parallel trends among people aged 15–54 years. (**C**) The AAPC was not significantly different between women and men (7.10% vs. 6.30%); *p* = 0.28 indicating parallel trends among people aged 35–54 years. (**D**) The AAPC was not significantly different between women and men (7.92% vs. 7.32%); *p* = 0.61 indicating parallel trends among people aged 15–34 years.

**Table 1 cancers-16-00971-t001:** Sex-specific trends for pancreatic ductal adenocarcinoma (PDAC) stratified by age groups.

		Trends					Comparison between Sex-Specific Trends (*p*-Value) ^e^			
Age Group (Years)	Number of Cases (%)	Time Period	APC ^a^ (95% CI)	APC *p*-Value	AAPC ^a^ (95% CI)	AAPC *p*-Value	AAPC Difference ^b^ (95% CI)	AAPC Difference	Test of Coincidence ^c^	Test of Parallelism ^d^
All Ages. N = 694,661										
Men	350,616 (50.5%)	2001–2007	1.10 * (0.66–1.53)	<0.001	0.65 * (0.48–0.81)	<0.001	0.05 (−0.18–0.28)	0.65	<0.001	0.09
		2007–2018	0.40 * (0.25–0.56)	<0.001						
Women	344,045 (49.5%)	2001–2008	1.19 * (0.84–1.54)	<0.001	0.59 * (0.43–0.76)	<0.001				
		2008–2018	0.18 * (0.00–0.36)	0.05						
Age > and =55. N = 623,552										
Men	308,831 (49.5%)	2001–2007	1.14 * (0.69–1.60)	<0.001	0.73 * (0.56–0.91)	<0.001	0.15 (−0.10–0.40)	0.24	<0.001	0.001
		2007–2018	0.51 * (0.36–0.67)	<0.001						
Women	314,721 (50.5%)	2001–2008	1.15 * (0.76–1.55)	<0.001	0.58 * (0.40–0.76)	<0.001				
		2008–2018	0.19 (−0.02–0.39)	0.07						
Age 15–54. N = 71,083										
Men	41,775 (58.8%)	2001–2018	−0.09 (−0.32–0.14)	0.44	−0.09 (−0.32–0.14)	0.44	−0.71 * (−1.07–−0.34)	<0.001	<0.001	0.02
Women	29,308 (41.2%)	2001–2018	0.62 * (0.30–0.94)	0.001	0.62 * (0.30–0.94)	<0.001				
Age 35–54. N = 69,501										
Men	40,918 (58.9%)	2001–2006	1.10 (−0.07–2.30)	0.06	0.01 (−0.35–0.38)	0.95	−0.61 * (−1.07–−0.14)	0.01	<0.001	0.003
		2006–2018	−0.44 * (−0.74–−0.14)	0.008						
Women	28,583 (41.1%)	2001–2018	0.62 * (0.31–0.93)	0.001	0.62 * (0.31–0.93)	<0.001				
Age 15–34. N = 1582										
Men	857 (54.2%)	2001–2018	0.78 (−0.81–2.39)	0.32	0.78 (−0.81–2.39)	0.32	−2.59 (−9.28–4.11)	0.45	0.003	0.04
Women	725 (45.8%)	2001–2003	38.20 (−22.21–145.54)	0.25	3.36 (−2.97–10.11)	0.30				
		2003–2018	−0.56 (−2.42–1.33)	0.53						

^a^. APC and AAPC refer to “annual percentage change” and “average annual percentage change”, respectively. * Indicates a significant difference (*p* < 0.05). Trends were calculated using version 4.9 of the Joinpoint Regression Program (National Cancer Institute). Up to four joinpoints (five line segments) were allowed. The annual percentage change over the whole period (2001–2018) is equal to the average over all the subgroups. ^b^. Negative value indicates greater average APC in women. ^c^. Test of whether sex-specific trends were identical. A significant *p*-value (*p* < 0.05) indicates that the trends were not identical (i.e., they had different incidence rates and the coincidence was rejected). ^d^. Test of whether sex-specific trends were equal. A significant *p*-value (*p* < 0.05) indicates that the trends were not equal (i.e., they had different incidence rates and parallelism was rejected). ^e^. Multiple testing correction was applied with *p*-value cutoffs at 0.05 for all ages and the prespecified older and younger adult subgroups, and at 0.025 for the post hoc 35–54 and 15–34 year-old age subgroups.

**Table 2 cancers-16-00971-t002:** Sex-specific trends for pancreatic neuroendocrine tumor (PanNET) stratified by age groups.

		Trends					Comparison between Sex-Specific Trends (*p*-Value) ^e^			
Age Group (Years)	Number of Cases (%)	Time Period	APC ^a^ (95% CI)	APC *p*-Value	AAPC ^a^ (95% CI)	AAPC *p*-Value	AAPC Difference ^b^ (95% CI)	AAPC Difference	Test of Coincidence ^c^	Test of Parallelism ^d^
All ages. N = 42,806										
Men	23,409 (54.7%)	2001–2010	7.35 * (6.38–8.33)	<0.001	7.60 * (6.31–8.91)	<0.001	0.24 (−1.63–2.12)	0.80	<0.001	0.58
		2010–2013	16.41 * (8.65–24.84)	<0.05						
		2013–2018	3.08 * (1.76–4.41)	<0.001						
Women	19,397 (45.3%)	2001–2010	6.83 * (5.85–7.82)	<0.001	7.36 * (6.02–8.72)	<0.001				
		2010–2013	15.77 * (7.67–24.49)	0.001						
		2013–2018	3.53 * (2.15–4.93)	<0.001						
Age > and =55. N= 29,788										
Men	16,867 (56.6%)	2001–2009	6.95 * (6.01–7.91)	<0.001	8.00 * (7.21–8.80)	<0.001	0.17 (−1.87–2.22)	0.87	<0.001	0.01
		2009–2013	15.73 * (12.50–19.05)	<0.001						
		2013–2018	3.81 * (2.83–4.80)	<0.001						
Women	12,921 (43.4%)	2001–2010	7.61 * (6.18–9.06)	<0.001	7.83 * (5.97–9.74)	<0.001				
		2010–2013	14.56 * (3.54–26.77)	<0.05						
		2013–2018	4.38 * (2.50–6.29)	<0.001						
Age 15–54. N= 12,962										
Men	6515 (50.3%)	2001–2016	8.00 * (7.08–8.93)	<0.001	6.36 * (4.46–8.28)	<0.001	−0.10 (−2.66–2.48)	0.95	0.63	0.72
		2016–2018	−5.23 (−18.86–10.68)	0.47						
Women	6447 (49.7%)	2001–2009	5.27 * (3.15–7.44)	<0.001	6.44 * (4.74–8.18)	<0.001				
		2009–2014	13.40 * (8.53–18.49)	<0.001						
		2014–2018	0.55 (−3.19–4.43)	0.75						
Age 35–54. N = 11,219										
Men	5763 (51.4%)	2001–2016	7.97 * (6.94–9.02)	<0.001	6.30 * (4.16–8.48)	<0.001	−0.80 (−4.23–2.63)	0.65	0.010	0.28
		2016–2018	−5.47 (−20.70–12.69)	0.50						
Women	5456 (48.6%)	2001–2003	15.48 (−5.00–40.36)	0.13	7.10 * (4.47–9.80)	<0.001				
		2003–2009	3.43 (−0.22–7.21)	0.06						
		2009–2014	14.17 * (9.60–18.94)	<0.001						
		2014–2018	0.34 (−3.15–3.95)	0.83						
Age 15–34. N = 1743										
Men	752 (43.1%)	2001–2018	7.32 * (5.82–8.83)	<0.001	7.32 * (5.82–8.83)	<0.001	−0.60 (−2.85–1.64)	0.60	<0.001	0.61
Women	991 (56.9%)	2001–2018	7.92 * (6.03–9.84)	<0.001	7.92 * (6.03–9.84)	<0.001				

^a^. APC and AAPC refer to “annual percentage change” and “average annual percentage change”, respectively. * Indicates a significant difference (*p* < 0.05). Trends were calculated using version 4.9 of the Joinpoint Regression Program (National Cancer Institute). Up to four joinpoints (five line segments) were allowed. The annual percentage change over the whole period (2001–2018) is equal to the average over all the subgroups. ^b^. Negative value indicates greater average APC in women. ^c^. Test of whether sex-specific trends were identical. A significant *p*-value (*p* < 0.05) indicates that the trends were not identical (i.e., they had different incidence rates and coincidence was rejected). ^d^. Test of whether sex-specific trends were equal. A significant *p*-value (*p* < 0.05) indicates that the trends were not equal (i.e., they had different incidence rates and parallelism was rejected). ^e^. Multiple testing correction was applied with *p*-value cutoffs at 0.05 for all ages and the prespecified older and younger adult subgroups, and at 0.025 for the post hoc 35–54 and 15–34 year-old age subgroups.

## Data Availability

Publicly available datasets were analyzed in this study. These data can be found here: https://www.cdc.gov/cancer/uscs/public-use/, accessed on 23 October 2021.

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
