# Peer review of "The Disproportionate Rise in Pancreatic Cancer in Younger Women Is Due to a Rise in Adenocarcinoma and Not Neuroendocrine Tumors: A Nationwide Time-Trend Analysis Using 2001–2018 United States Cancer Statistics Databases"

_cancers, 2024, doi:10.3390/cancers16050971_

Round 1

Reviewer 1 Report

Comments and Suggestions for Authors

The  authors used the SEER data on USA population during the time of 2000-2019 and show an increase in incidence in PDAC, mainly in the women.

PNET also increase, but this was not sex-dependent.

This data shows interesting changes in incidence over time.

The paper could be improved by:

- more legible figures which show the changes more eloquent

- more discussion on why these changes happen

- is there a correlation with different radiologic modalities used over the years? More CAT scans during the years from 2010?

- add figure on correlation to ethnicity and incidence

- is there correlation to regions giving clues as which risk factors could induce the increase in incidence? higher obesity in certain states?

Author Response

Point-by-point response to comments from Reviewer 1

Comments 1: more legible figures which show the changes more eloquent

Response 1: Thank you for your suggestion. We have added more figures and made the changes to the current updated version of the manuscript. (page 3, line 104; page 4, line 121; page 7, lines 145-153; page 8, lines 154-160).

Comments 2: more discussion on why these changes happen

Response 2:

Thank you for your inquiry. In the revised manuscript, we have expanded the discussion to focus on several key topics.

1. Explanations for the sex disparity observed in young adults with PDAC. While this observational study cannot yield definitive conclusions due to the absence of variables pertaining to risk factors in the database, we conducted an extensive literature review. Imbalanced PDAC-exclusive risk factors and genetic susceptibility are suspected contributors. Our updated manuscript discusses the nuanced discussion surrounding patterns of tobacco use and heavy alcohol consumption, as well as reproductive factors influenced by estrogen-related genetic polymorphisms.

2. Insights into the consistent increase in PanNET incidence without sex disparity. Some literature suggests that the enhanced utilization of imaging and endoscopy modalities, leading to the detection of asymptomatic patients, may contribute to this trend. Additionally, while both PDAC and PanNET are associated with risk factors such as tobacco use, alcohol consumption, and metabolic syndrome, we suspect that the patterns of impact for these risk factors are different. Further investigations are needed to elucidate the underlying details.

Please refer to the revised discussion and updated references provided in manuscript (in red). (Discussion section, page 8, lines 176-205; page 9, lines 206-210, and lines 220-222. Reference section, page 10-12, lines 275-360)

Comments 3: is there a correlation with different radiologic modalities used over the years? More CAT scans during the years from 2010?

Response 3:  Thank you for these very important suggestions.

The rising incidence of pancreatic cancer among younger adults has been documented in studies utilizing various databases in the U.S. [1]. These trends appear to parallel the increasing incidence of other young-onset gastrointestinal (GI) cancers, such as colorectal and gallbladder malignancies [2, 3].

Improvement in diagnostics and data recording in many countries and regions has been suggested as contributing factors to the overall increase, at least partially, in pancreatic cancer and early-onset GI malignancies [4]. A retrospective cohort study examining patterns of medical imaging between 2000 and 2016 across seven integrated U.S. healthcare systems and Ontario, Canada, revealed a continued rise in the utilization of computed tomography (CT) and magnetic resonance imaging (MRI) among adults [5]. Particularly, the utilization of abdominal imaging has markedly increased, as indicated by database analysis from the Medicare fee-for-service population [6]. Similarly, CT utilization has consistently increased in emergency department settings, with abdominal CT scans accounting for over 50% of all CT procedures performed [7]. Additionally, the use of endoscopic ultrasound has also increased significantly, as reported in a study using SEER-Medicare database [8]. Similar trends have been found in other regions globally [9, 10].

Differences in patterns of healthcare utilization, coupled with evolving awareness among healthcare providers regarding early-onset GI cancers, have been suspected as potential contributors to sex disparities among young adults with pancreatic cancer [4]. However, it is important to recognize that multiple layers of imbalanced risk factors contribute to complex interactions involving race, sex, age, and genetic predisposition. A deeper understanding of the underlying pathogenesis across various population groups is crucial for investigating and addressing these disparities.

Comments 4: add figure on correlation to ethnicity and incidence

Response 4:  We agreed that the evaluation of race- and ethnicity-specific incidence trends of pancreatic cancer will provide crucial insights into the influence of demographics, psychosocial, societal, and systemic factors on pancreatic cancer incidence.

Our research group has published two studies to this field. The first study, based on SEER 21 data [11] analyzed the adjusted incidence rates (aIR) for pancreatic cancer between 2000 and 2018, organizing race and ethnicity into Non-Hispanic White (White), Non-Hispanic Black (Black) and Hispanic groups. We found that younger women of all races and ethnicities demonstrated a greater rate of increase in aIR compared to the counterpart men. Notably, younger Hispanic women exhibited a significantly higher rate of increase in aIR compared to younger Black women [average annual percentage change (AAPC) difference = −1.28, p = 0.028)] and younger White women [AAPC difference = −1.35, p = 0.011)]. The second study based on 2001-2018 National Program of Cancer Registries (NPCR)-based [1] categorized race into Black, White, and other groups. Our findings revealed that although White women experienced increasing aIRs at a greater rate compared with men (41,686 cases; 41.87% women; AAPC difference, 1.59%), a more dramatic increase was seen in women of Black race compared with counterpart men (9498 cases; 46.60% women; AAPC difference, 2.23%).

Another study conducted by Huang et al. [12] utilizing data from the North American Association of Central Cancer Registries (NAACCR), focused on the racial and ethnic disparities in patients diagnosed with pancreatic ductal adenocarcinoma (PDAC) between 1995 and 2018. The study identified that the increasing incidence of early-onset PDAC in women was primarily driven by non-Hispanic Whites [AAPC 0.99; 95% confidential interval(CI), 0.73-1.25] and Hispanics (AAPC 0.68; 95% CI, 0.17-1.20), consistent with previous SEER-based analyses [13].

Given the comprehensive investigation of the impact of race and ethnicity on pancreatic cancer incidence through publications from our group and other researchers, we have not included additional information in the current manuscript. The trend figures referenced in this paragraph can be accessed in the respective articles mentioned.

Comments 5: is there correlation to regions giving clues as which risk factors could induce the increase in incidence? higher obesity in certain states?

Response 5:  Thank you for raising this intriguing question. Unfortunately, due to limitations in the USCS database, we were unable to access information on major modifiable risk factors, including obesity, smoking, and alcohol consumption, as well as their sex-specific, age-specific, or region-specific effects. This limitation has been acknowledged and updated in discussion section of the main manuscript (page 9, discussion section, lines 220-222).

However, we remain optimistic that our study will contribute valuable insights to inform health policies aimed at identifying individuals at risk and laying the groundwork for future prospective studies. Ideally, a longitudinal cohort study could be conducted, incorporating age-period modeling to distinguish cohort effects from period effects. This study could account for various risk factors such as BMI at different ages, waist circumference, markers for metabolic syndrome (hypertension, diabetes, hyperlipidemia), dietary factors (healthy eating index score), smoking and alcohol use status, socioeconomic status, physical activity, women’s health-related factors (such as pre/post-menopausal status, oral contraception use, history of deliveries, hormonal replacement therapy, etc.), and genetic polymorphisms.

References

[1] Abboud, Y.; Samaan, J. S.; Oh, J.; Jiang, Y.; Randhawa, N.; Lew, D.; Ghaith, J.; Pala, P.; Leyson, C.; Watson, R.; et al. Increasing Pancreatic Cancer Incidence in Young Women in the United States: A Population-Based Time-Trend Analysis, 2001-2018. Gastroenterology 2023, 164 (6), 978-989.e976. DOI: 10.1053/j.gastro.2023.01.022.

[2] Sung, H.; Siegel, R. L.; Rosenberg, P. S.; Jemal, A. Emerging cancer trends among young adults in the USA: analysis of a population-based cancer registry. Lancet Public Health 2019, 4 (3), e137-e147. DOI: 10.1016/S2468-2667(18)30267-6.

[3] Patel, S. G.; Karlitz, J. J.; Yen, T.; Lieu, C. H.; Boland, C. R. The rising tide of early-onset colorectal cancer: a comprehensive review of epidemiology, clinical features, biology, risk factors, prevention, and early detection. Lancet Gastroenterol Hepatol 2022, 7 (3), 262-274. DOI: 10.1016/S2468-1253(21)00426-X.

[4] The Lancet Gastroenterology Hepatology. Cause for concern: the rising incidence of early-onset pancreatic cancer. Lancet Gastroenterol Hepatol 2023, 8 (4), 287. DOI: 10.1016/S2468-1253(23)00039-0.

[5] Smith-Bindman, R.; Kwan, M. L.; Marlow, E. C.; Theis, M. K.; Bolch, W.; Cheng, S. Y.; Bowles, E. J. A.; Duncan, J. R.; Greenlee, R. T.; Kushi, L. H.; et al. Trends in Use of Medical Imaging in US Health Care Systems and in Ontario, Canada, 2000-2016. JAMA 2019, 322 (9), 843-856. DOI: 10.1001/jama.2019.11456.

[6] Moreno, C. C.; Hemingway, J.; Johnson, A. C.; Hughes, D. R.; Mittal, P. K.; Duszak, R. Changing Abdominal Imaging Utilization Patterns: Perspectives From Medicare Beneficiaries Over Two Decades. J Am Coll Radiol 2016, 13 (8), 894-903. DOI: 10.1016/j.jacr.2016.02.031.

[7] Hess, E. P.; Haas, L. R.; Shah, N. D.; Stroebel, R. J.; Denham, C. R.; Swensen, S. J. Trends in computed tomography utilization rates: a longitudinal practice-based study. J Patient Saf 2014, 10 (1), 52-58. DOI: 10.1097/PTS.0b013e3182948b1a.

[8] Huntington, C. R.; Walsh, K.; Han, Y.; Salo, J.; Hill, J. National Trends in Utilization of Endoscopic Ultrasound for Gastric Cancer: a SEER-Medicare Study. J Gastrointest Surg 2016, 20 (1), 154-163; discussion 163-154. DOI: 10.1007/s11605-015-2988-8.

[9] Ha, T. N.; Kamarova, S.; Youens, D.; Wright, C.; McRobbie, D.; Doust, J.; Slavotinek, J.; Bulsara, M. K.; Moorin, R. Trend in CT utilisation and its impact on length of stay, readmission and hospital mortality in Western Australia tertiary hospitals: an analysis of linked administrative data 2003-2015. BMJ Open 2022, 12 (6), e059242. DOI: 10.1136/bmjopen-2021-059242.

[10] Leeds, J.; Awadelkarim, B.; Lam, K.; Maheshwari, P.; Nayar, M.; Johnson, S.; Oppong, K. O39 Trends in pancreatic cystic lesions undergoing endoscopic ultrasound: 16 years experience in a tertiary center. The BSG (British Society of Gastroenterology) Annual meeting, London, UK 2021.

[11] Samaan, J. S.; Abboud, Y.; Oh, J.; Jiang, Y.; Watson, R.; Park, K.; Liu, Q.; Atkins, K.; Hendifar, A.; Gong, J.; et al. Pancreatic Cancer Incidence Trends by Race, Ethnicity, Age and Sex in the United States: A Population-Based Study, 2000-2018. Cancers (Basel) 2023, 15 (3). DOI: 10.3390/cancers15030870.

[12] Huang, B. Z.; Liu, L.; Zhang, J.; Pandol, S. J.; Grossman, S. R.; Setiawan, V. W.; Team, U. P. R. Rising Incidence and Racial Disparities of Early-Onset Pancreatic Cancer in the United States, 1995-2018. Gastroenterology 2022, 163 (1), 310-312.e311. DOI: 10.1053/j.gastro.2022.03.011.

[13] Gordon-Dseagu, V. L.; Devesa, S. S.; Goggins, M.; Stolzenberg-Solomon, R. Pancreatic cancer incidence trends: evidence from the Surveillance, Epidemiology and End Results (SEER) population-based data. Int J Epidemiol 2018, 47 (2), 427-439. DOI: 10.1093/ije/dyx232.

Reviewer 2 Report

Comments and Suggestions for Authors

This study aimed to perform age and sex-specific time-trend analysis of age-adjusted incidence rates (aIRs) of PDAC and PanNET using the United States Cancer Statistics (USCS) database, which represents nearly 100% of the US population. The communication is interesting and provide important data. The novelty and significance of this communication is high considering clinical aspects, and giving rise to future research hypotheses.

I have only minor, technical comments:

1. Please adjust tables and their content for better eligibility.

2. The Authors should consider to mark p values of specific comparison in red, as long as these p values indicate statistically significant data.

Author Response

Comments 1: Please adjust tables and their content for better eligibility.

Response 1: Thank you for your suggestions. We have adjusted the table and the content for better readability and eligibility. Changes have been made to the current updated version of the manuscript in pages 3-4, lines 105-106, table 1 and pages 4-6, lines 122-131, table 2.

If the article is accepted, we would like to respectfully discuss with the editors the potential option of incorporating a landscape view, taking into account whether it aligns with the journal's formatting guidelines. We recognize the significant volume of data involved and believe that this consideration may contribute to enhancing the presentation of our research findings.

Comments 2: The Authors should consider to mark p values of specific comparison in red, as long as these p values indicate statistically significant data.

Response 2: Thank you for the insightful suggestion. We have marked p-values of specific comparisons in red to indicate statistically significant data, aiming to enhance the clarity and readability of our results. Shown in pages 3 of 12, lines 99-102; pages 3-4, lines 105-106, table 1 and pages 4-6, lines 122-131, table 2.